# Prevalence and spatial distribution characteristics of human echinococcosis in China

**Li-Ying Wang**[1,2,3,4], **Min Qin**[1], **Ze-Hang Liu**[1], **Wei-Ping Wu**[1], **Ning Xiao**[1], **Xiao-Nong Zhou**[1]\*, **Sylvie Manguin**[4], **Laurent Gavotte**[2], **Roger Frutos**[3]

**1** National Institute of Parasitic Diseases, Chinese Centre for Disease Control and Prevention (Chinese Centre for Tropical Diseases Research); NHC Key Laboratory of Parasite and Vector Biology; WHO Collaborating Centre for Tropical Diseases; National Centre for International Research on Tropical Diseases, Shanghai, China, **2** Espace-Dev, UMR D-228, Université de Montpellier, Montpellier, France, **3** Cirad, UMR 17, Intertryp, Campus international de Baillarguet, Montpellier, France, **4** HydroSciences Montpellier (HSM), Institut de Recherche pour le Développement (IRD), CNRS, Université Montpellier, Montpellier, France

\* zhouxn1@chinacdc.cn

**Data Availability Statement:** All relevant data are within the manuscript.

## Abstract

### Background

Echinococcosis is a zoonotic parasitic disease caused by larval stages of cestodes belonging to the genus *Echinococcus*. The infection affects people's health and safety as well as agropastoral sector. In China, human echinococcosis is a major public health burden, especially in western China. Echinococcosis affects people health as well as agricultural and pastoral economy. Therefore, it is important to understand the prevalence status and spatial distribution of human echinococcosis in order to advance our knowledge of basic information for prevention and control measures reinforcement.

### Methods

Report data on echinococcosis were collected in 370 counties in China in 2018 and were used to assess prevalence and spatial distribution. SPSS 21.0 was used to obtain the prevalence rate for CE and AE. For statistical analyses and mapping, all data were processed using SPSS 21.0 and ArcGIS 10.4, respectively. Chi-square test and Exact probability method were used to assess spatial autocorrelation and spatial clustering.

### Results

A total of 47,278 cases of echinococcosis were recorded in 2018 in 370 endemic counties in China. The prevalence rate of human echinococcosis was 10.57 per 10,000. Analysis of the disease prevalence showed obvious spatial positive autocorrelation in globle spatial autocorrelation with two aggregation modes in local spatial autocorrelation, namely high-high and low-high aggregation areas. The high-high gathering areas were mainly concentrated in northern Tibet, western Qinghai, and Ganzi in the Tibetan Autonomous Region and in

**Funding:** This work was supported by the National Natural Science Foundation of China (http://www.nsfc.gov.cn/) [Grant No. 81703281]. The funders had no role in study design, data collection and analysis, decision to publish, or preparation of the manuscript.

**Competing interests:** The authors have declared that no competing interests exist.

Sichuan. The low-high clusters were concentrated in Gamba, Kangma and Yadong counties of Tibet. In addition, spatial scanning analysis revealed two spatial clusters. One type of spatial clusters included 71 counties in Tibet Autonomous Region, 22 counties in Qinghai, 11 counties in Sichuan, three counties in Xinjiang Uygur Autonomous Region, two counties in Yunnan, and one county in Gansu. In the second category, six types of spatial clusters were observed in the counties of Xinjiang Uygur Autonomous Region, and the Qinghai, Gansu, and Sichuan Provinces.

## Conclusion

This study showed a serious prevalence of human echinococcosis with obvious spatial aggregation of the disease prevalence in China. The Qinghai-Tibet Plateau is the "hot spot" area of human echinococcosis in China. Findings from this study indicate that there is an urgent need of joint strategies to strengthen efforts for the prevention and control of echinococcosis in China, especially in the Qinghai-Tibet Plateau.

### Author summary

Echinococcosis is a zoonotic parasitic disease caused by larval stages of cestodes belonging to the genus *Echinococcus*. In China, human echinococcosis is a major public health burden, especially in western China. Therefore, it is important to understand the prevalence status and spatial distribution of human echinococcosis in order to provide basic information for prevention and control measures reinforcement. To describe the distribution and analyze the prevalence and spatial distribution characteristics of human echinococcosis in China, report data of echinococcosis were collected in 370 counties in 2018. For the year 2018, there were 47,278 cases of echinococcosis recorded in 370 endemic counties in China. Analysis of the disease prevalence showed obvious spatial positive autocorrelation in global spatial autocorrelation with two aggregation modes in local spatial autocorrelation, namely high-high and low-high aggregation areas. The high-high gathering areas were mainly concentrated in northern Tibet, western Qinghai, and Ganzi in the Tibetan Autonomous Region and in Sichuan. This study showed obvious spatial aggregation of human echinococcosis prevalence in China. The Qinghai-Tibet Plateau is the "hot spot" area of human echinococcosis in China. Such findings indicate that here is an urgent need of joint strategies to strengthen efforts for the prevention and control of echinococcosis in China, especially in the Qinghai-Tibet Plateau.

## 1. Introduction

Echinococcosis is a zoonotic parasitic disease caused by larval stages of cestodes belonging to the genus *Echinococcus* and present worldwide. The life-cycle of the echinococcosis parasites involves carnivores as definitive hosts which harbour the adult egg-producing stage in the intestine and intermediate host animals in which the infective metacestode stage develops after peroral infection with eggs. In China, two forms of human echinococcosis are found: cystic echinococcosis (CE) caused by the larvae of *Echinococcus granulosus* and alveolar echinococcosis (AE) caused by the larvae of *Echinococcus multilocularis* [1].

China reported the highest prevalence rate for human echinococcosis in the world, with a disease burden estimated at 322,400 disability-adjusted life years (DALYs) [2]. In 2017, the Chinese DALYs of echinococcosis has been estimated at 293,400 years lost to disability (YLD) and 28,800 years of life lost (YLL) [2]. In western China, about nine million people in six provinces are under risk of AE [3]. Both CE and AE are major public health issues worldwide [4]. They heavily impair the patients, especially AE, with a mortality rate of about 90% in the past ten years if the patients untreated or treated inadequately [5]. A national survey of echinococcosis infection conducted conducted from 2012 to 2016 showed that 368 out of 413 counties were endemic. An overall detection rate of 0.46% was found over 364 endemic counties from nine provinces and autonomous regions including the Autonomous Region of Tibet (usually referred to as "Tibet") Sichuan, Qinghai, Xinjiang Uygur Autonomous Region, Gansu, Ningxia, Inner Mongolia, Yunnan, and Shaanxi. Tibet dislayed the highest scores with a detection rate of 1.71% and an estimated prevalence of 1.66% [6]. Currently, 370 counties are endemic after detection of the disease in Dongxiang County, Gansu Province and in Ulagai Management District, Inner Mongolia in 2017. Western China is known as the world highest endemic area for both CE and AE [7], making echinococcosis a public health priority in this region [4,8].

However, this burden is not well assessed yet in terms of accurate spatial distribution, patterns, and clusters which are of capital importance in public health for effective disease control strategies implementation. Spatial analyses, and in particular assessment of clusters and spatial aggregation, are therefore research priorities [9–12]. This work was conducted in order to prioritize and optimize control actions against human echinococcosis, with an overall objective to understand the prevalence, spatial distribution and dynamic of echinococcosis within the Chinese population by the end of 2018.

## 2. Methods

### 2.1. Ethics statement

This survey consisted of the collection of report data of echinococcosis cases in 370 counties in 2018 was approved by the Ethics Review Committee of the National Institute of Parasitic Diseases, Chinese Center for Disease Control and Prevention (No. 20160810). All participants were informed about the content and purpose of the investigation and examination, complications, consequences and benefits before data collection. The consenters were required to sign the "informed consent form". These activities are all within the scope of the national project for echinococcosis control.

### 2.2. Source of data

Demographic data relating to the population exposed to human echinococcosis in the 370 endemic counties were obtained from the population survey released by the Bureau of Statistics. The list of clinically diagnosed and confirmed cases of human echinococcosis comes from the official national annual report on echinococcosis. Data were statistically analyzed by county. Data from the Xinjiang Production and Construction Corps were considered at the division level. However, no spatial data were available. Therefore, these data were associated with the corresponding counties from the Xinjiang Autonomous Region in order to perform spatial autocorrelation and aggregation analyses.

### 2.3. Classification of prevalence

In order to distinguish the prevalence of newly diagnosed cases from previously existing cases of human echinococcosis in the 370 endemic counties of China, a descriptive statistical

analysis was performed using the SPSS 21.0 software package (IBM, Armonk, USA). A statistical classification was also carried out by type of case. Prevalence of human echinococcosis was classified according to the classification standards for endemic counties as reported in the 2019 edition of technical guidelines for echinococcosis control [13]. Considering the differences between CE and AE in transmission cycle, preventive strategies, control measures, clinical manifestations and treatment regimens, we described and analyzed the characteristics of CE and AE independently.

## 2.4. Spatial autocorrelation

Spatial autocorrelation analyses were condueted to measure the degree of aggregation of spatial unit attribute values and to determine whether a variable is spatially correlated and relevant [14]. Global and local spatial autocorrelations were performed. Global spatial autocorrelation is meant to assess the patterns of spatial aggregation over a whole area. The Moran's $I$ index was used to express the global spatial autocorrelation. The value of the Moran's $I$ index ranges from -1 to 1. A null value indicates a lack of correlation within the area considered. A positive value indicates a positive spatial correlation of the index whereas a negative value indicates a negative spatial correlation. The higher the absolute value, the stronger the correlation. However, global correlation analysis ignores the existence of spatial heterogeneity, and can only be used to measure the overall correlation instead of the spatial distribution within a given area. The local spatial autocorrelation was thus calculated to evaluate the correlation between each spatial unit and surrounding areas, effectively expressing the heterogeneity and homogeneity of data. An aggregation pattern can be identified in the combination of the Moran scatter plot and the local Moran's $I$ test. The local spatial autocorrelation can be divided into four categories: high-high, low-low, high-low, and low-high. High-high and low-low patterns denote a strong positive spatial correlation of the observations whereas high-low and low-high patterns denote a strong negative spatial correlation [15]. The Moran's $I$ values of global and local spatial autocorrelations were calculated using the spatial statistical analysis module of ArcGIS version 10.4 (Esri Inc., Redlands, CA, USA), with local indicators of spatial association (LISA) cluster map for visualization.

## 2.5. Spatial scan clustering

The spatial scan clustering is a method for analyzing data based on a moving scanning window. This analysis can directly show the distribution of diseases and model the trend of diseases expansion [10,16,17]. Areas of high incidence were scanned using a moving circular window dynamically varying in size. A retrospective spatial scan analysis was performed using SaTScan V9.5 (Management Information Services, Maryland, USA). The number of Monte Carlo randomization tests was 999 and the maximum spatial scan area was set to 25% of the total population. Log-likelihood ratio (LLR) under the dynamically varying window was calculated to determine potential clusters. Finally, the window with the highest LLR value was defined as the most likely cluster. Other clusters displaying statistically significant LLRs were defined as secondary clusters. Results were visualized using Arcgis10.4.

# 3. Results

## 3.1. Prevalence of human echinococcosis

In 2018, a total number of 47,278 cases of human echinococcosis were recorded in the 370 epidemic counties. The endangered population was estimated at 44,730,268 with a prevalence rate of 10.57 per 10,000 (Table 1). Out of these, 33,578 (71.02%) were CE cases and 10,398

Table 1. Types and prevalence status of human echinococcosis in China, 2018.

| Province/ Autonomous region | Number of endemic counties for monoinfection | Population of endemic areas | Number of endemic counties for mixed CE and AE | Population of endemic areas for mixed CE and AE | All cases | | | | | Prevalence rate (1/10000) | Prevalence rate of CE (1/10000) | Prevalence rate of AE (1/10000) | Cases found in 2018 | | | | | |
|---|---|---|---|---|---|---|---|---|---|---|---|---|---|---|---|---|---|---|
| | | | | | Total cases | CE | AE | Mixed CE and AE cases | Unclassified cases | | | | Number of new cases | Incidence rate (1/10000) | CE | AE | Mixed CE and AE cases | Unclassified cases |
| Inner Mongolia | 26 | 1800838 | 0 | 0 | 124 | 123 | 0 | 0 | 1 | 0.69 | 0.68 | / | 31 | 0.17 | 30 | 0 | 0 | 1 |
| Sichuan | 35 | 1357646 | 11 | 641292 | 12291 | 5993 | 5314 | 96 | 888 | 90.53 | 44.85 | 84.36 | 318 | 2.34 | 215 | 90 | 8 | 5 |
| Yunnan | 24 | 1541610 | 0 | 0 | 37 | 36 | 0 | 0 | 1 | 0.24 | 0.23 | / | 10 | 0.06 | 9 | 0 | 0 | 1 |
| Tibet | 74 | 2704690 | 47 | 1832317 | 14983 | 13854 | 668 | 53 | 408 | 55.4 | 51.42 | 3.93 | 1048 | 3.87 | 966 | 48 | 22 | 12 |
| Shaanxi | 2 | 430615 | 0 | 0 | 77 | 77 | 0 | 0 | 0 | 1.79 | 1.79 | / | 2 | 0.05 | 2 | 0 | 0 | 0 |
| Gansu | 57 | 11288497 | 10 | 2594440 | 1749 | 1695 | 54 | 0 | 0 | 1.55 | 1.50 | 0.21 | 300 | 0.27 | 299 | 1 | 0 | 0 |
| Qinghai | 39 | 4882008 | 14 | 782728 | 12513 | 6629 | 4116 | 134 | 1634 | 25.63 | 13.85 | 54.30 | 649 | 1.33 | 346 | 124 | 5 | 174 |
| Ningxia | 19 | 3513401 | 3 | 1055701 | 1903 | 1711 | 176 | 16 | 0 | 5.42 | 4.92 | 1.82 | 119 | 0.34 | 111 | 7 | 1 | 0 |
| Xinjiang | 81 | 16031078 | 30 | 5734874 | 3463 | 3332 | 65 | 8 | 58 | 2.16 | 2.08 | 0.13 | 1220 | 0.76 | 1147 | 15 | 4 | 54 |
| The Xinjiang Production and Construction Crops | 13 | 1179885 | 0 | 0 | 138 | 128 | 5 | 0 | 5 | 1.17 | 1.08 | / | 111 | 0.94 | 101 | 5 | 0 | 5 |
| Total | 370 | 44730268 | 115 | 12641352 | 47278 | 33578 | 10398 | 307 | 2995 | 10.57 | 7.58 | 8.47 | 3808 | 0.85 | 3226 | 290 | 40 | 252 |

(21.99%) were AE cases. Mixed CE and AE infections represented 307 cases (0.65%) while 2,995 cases (6.34%) could not be clearly classified (either CE or AE). This lack of specific identification in the records was due to i) the absence of notification, ii) the absence of clear description of the cystic or vesicular status, and iii) the absence of B-ultrasound pictures making it impossible to screen and reclassify (Table 1). Local cases of CE were found in each of the 370 endemic counties, with 12,641,352 people at risk and a prevalence rate of 7.58 per 10,000. A total of 115 counties have been previously identified as AE epidemic areas in the 2012–2016 national echinococcosis survey. We found in this study a prevalence rate of 8.47 per 10,000. Notably, the prevalence of AE was higher than that of CE (Table 1). This did not include the unclassified cases leading thus to a slightly underestimated prevalence. From these, 3,808 new cases were reported in 2018, accounting for 8.05% of the total cases. The annual prevalence of human echinococcosis in 2018 was 0.85 per 10,000 (Table 1). The annual prevalence rate of CE was 0.73 per 10,000, and that of AE was 0.26 per 10,000.

### 3.2. Classification of the prevalence of human echinococcosis in China

In this study, we found that the prevalence varied significantly among provinces and even between endemic counties within the same province. In the National Echinococcosis Prevention and Control technical plan, the epidemic degree was classified according to the prevalence rate on human and the infection rate on dogs. Here, we only addressed the classification criteria of human prevalence. Class I epidemic counties correspond to prevalence rates higher than or equal to 100/10,000; Class II epidemic counties display prevalence rates comprised between 10/10,000 and 100/10,000; Class III epidemic counties are characterized by prevalence rates ranging between 0 and 10/10,000; Class IV corresponds to counties free of human echinococcosis cases. We calculated and classified the prevalence of echinococcosis for each county. Twenty nine class I epidemic counties (7.84%) (Table 2) were recorded. The highest prevalence was found in the counties of Shiqu (642.07/10,000), Gadee (461.04/10,000), and Tarlag (424.84/10,000) (Fig 1). A total of 96 type II counties (25.95%) and 210 type III counties (56.76%). A total of were recorded while 35 counties were classified as class IV with no cases of human echinococcosis reported in 2018. (Table 2). CE and AE were analyzed and classified

**Table 2. Classification of the prevalence rate for human echinococcosis in China, 2018.**

| Province/ Autonomous region | Total number of. counties | $P \geq 100/10000$ | | $10/10000 \leq P < 100/10000$ | | $0 < P < 10/10000$ | | $P = 0$ | |
|---|---|---|---|---|---|---|---|---|---|
| | | Number of counties | constituent ratio (%) | Number of counties | Constituent ratio (%) | Number of counties | Constituent ratio (%) | Nunmber of counties | Constituent ratio (%) |
| Inner Mongolia | 26 | 0 | 0 | 1 | 3.85 | 12 | 46.15 | 13 | 50 |
| Sichuan | 35 | 6 | 17.14 | 14 | 40 | 15 | 42.86 | 0 | 0 |
| Yunnan | 24 | 0 | 0 | 0 | 0 | 11 | 45.83 | 13 | 54.17 |
| Tibet | 74 | 16 | 21.62 | 51 | 68.92 | 7 | 9.46 | 0 | 0 |
| Shaanxi | 2 | 0 | 0 | 0 | 0 | 1 | 50 | 1 | 50 |
| Gansu | 57 | 0 | 0 | 6 | 10.53 | 47 | 82.46 | 4 | 7.02 |
| Qinghai | 39 | 7 | 17.95 | 15 | 38.46 | 17 | 43.59 | 0 | 0 |
| Ningxia | 19 | 0 | 0 | 3 | 15.79 | 16 | 84.21 | 0 | 0 |
| Xinjiang | 81 | 0 | 0 | 6 | 7.41 | 72 | 88.89 | 3 | 3.7 |
| Xinjiang Production and Construction Crops | 13 | 0 | 0 | 0 | 0 | 12 | 92.31 | 1 | 7.69 |
| Total | 370 | 29 | 7.84 | 96 | 25.95 | 210 | 56.76 | 35 | 9.46 |

*P*: prevalence rate

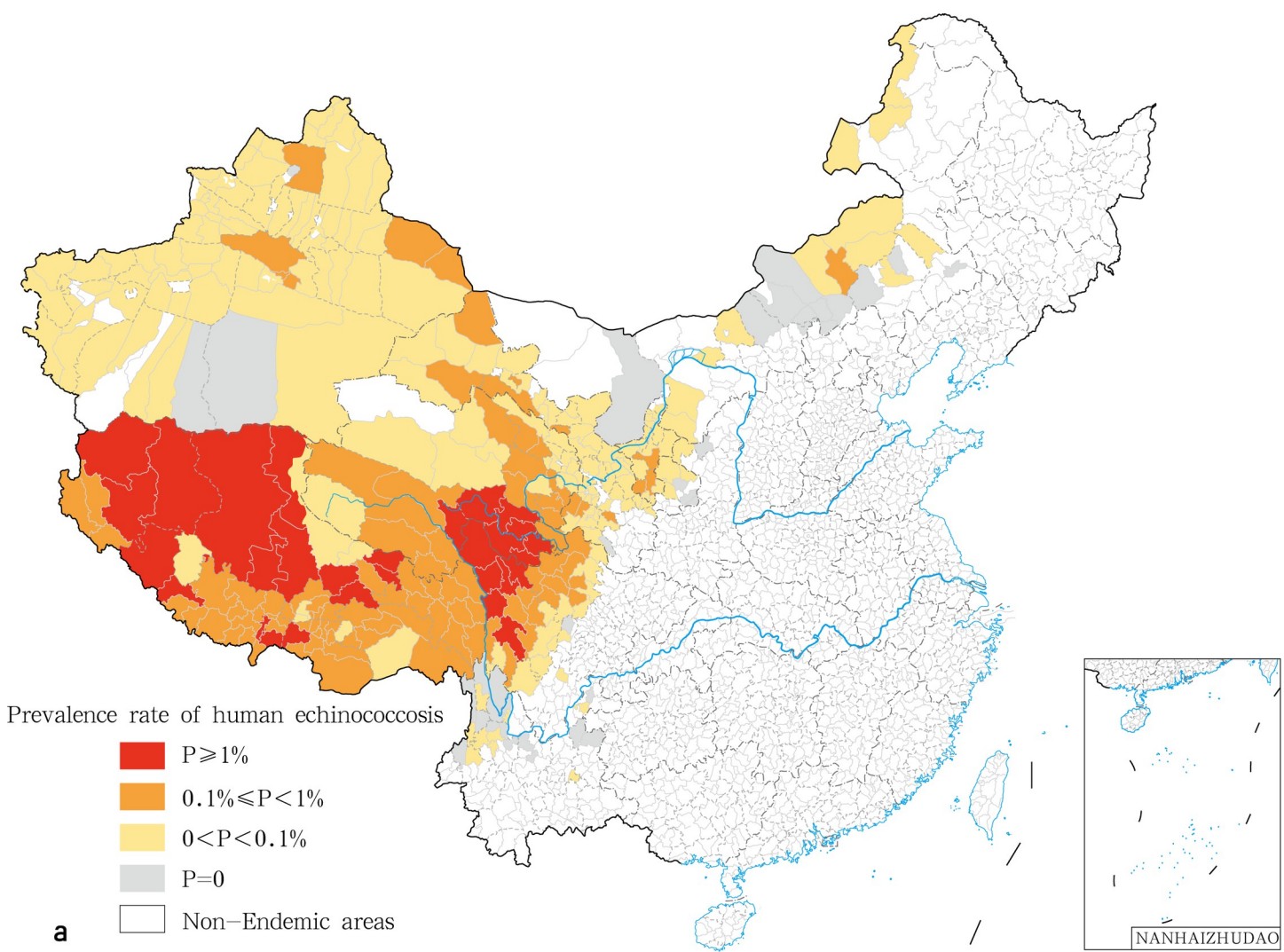

**Fig 1. The spatial distribution of human echinococcosis in China in 2018.** The base layer is from https://www.webmap.cn/mapDataAction.do?method=forw&resType=5&storeId=2&storeName=%E5%9B%BD%E5%AE%B6%E5%9F%BA%E7%A1%80%E5%9C%B0%E7%90%86%E4%BF%A1%E6%81%AF%E4%B8%AD%E5%BF%83&fileId=BA420C422A254198BAA5ABAB9CAAFBC1 with credit to National Catalogue Service For Geographic Information.

independently. We found 21 class I counties for CE, mainly distributed in the central region of the Qinghai Tibet Plateau, including 14 counties in Tibet Autonomous Region, three counties in Sichuan (Shiqu, Seertar and Litang), and four counties in Qinghai (Tarlag, Jigzhi, Chindu and Gadee) (Fig 2). The counties with the highest incidence for CE were Gadee (301.1 / 10,000) in Qinghai, Zhongba county (255.1/ 10,000) and Baqeen (246.6/ 10,000) in Tibet, and Shiqu County (236.0 / 10,000) in Sichuan (Fig 2). Ninety eight Class II counties in seven provinces were recorded for CE (Fig 2). An additional 44 epidemic counties for CE displayed the conditions for transmission but with no local CE patients (Table 3 and Fig 2). With respect to AE, 6 class I counties were recorded, i.e. Shiqu (402.2 / 10,000) and Seertar (176.0 / 10,000) in Sichuan, Tarlag (291.3/ 10,000), Baima (227.9 / 10,000), Jigzhi (202.2/ 10,000) and Chido (161.0/10,000) in Qinghai (Fig 3). The Class II category comprised 10 counties (8.7%), including: Baiyu, Zamtang, Deegee and Garzee in Sichuan, Baqeen, Xainza and Bangoin in Tibet, Madoi, Gadee and Maqeen in Qinghai (Fig 3). Forty six counties from 6 provinces were

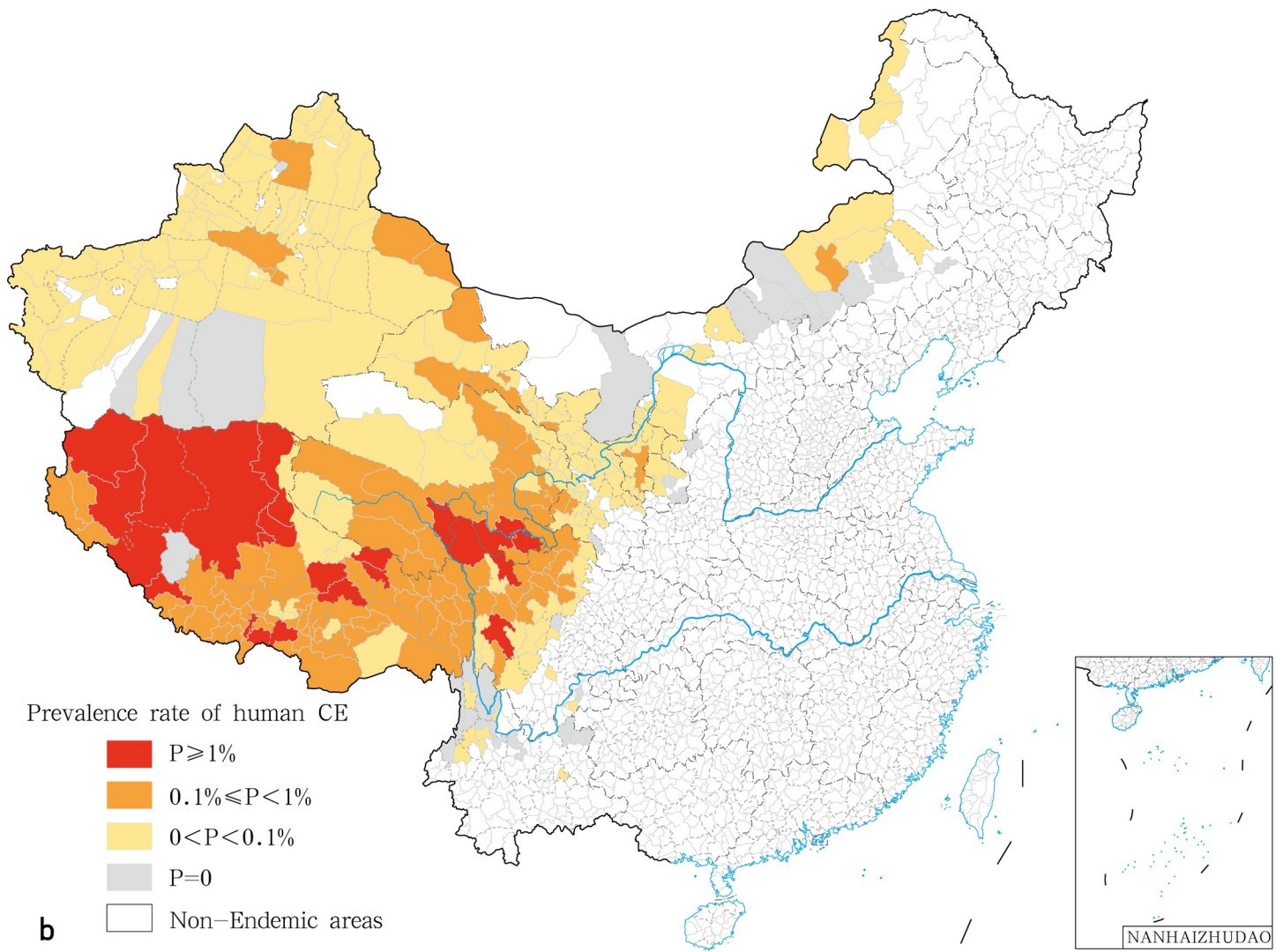

Prevalence rate of human CE

▮ P≥1%

▮ 0.1%≤P<1%

▮ 0<P<0.1%

▮ P=0

☐ Non−Endemic areas

b

**Fig 2. The spatial distribution of human CE in China in 2018.** The base layer is from https://www.webmap.cn/mapDataAction.do?method=forw&resType=5&storeId= 2&storeName=%E5%9B%BD%E5%AE%B6%E5%9F%BA%E7%A1%80%E5%9C%B0%E7%90%86%E4%BF%A1%E6%81%AF%E4%B8%AD%E5%BF%83&fileId= BA420C422A254198BAA5ABAB9CAAFBC1 with credit to National Catalogue Service For Geographic Information.

classified as Class III (40.0%). A total of 53 counties were categorized as class IV (46.1%) (Table 4 and Fig 3).

## 3.3. Spatial distribution of human echinococcosis in China

The global spatial autocorrelation analysis was performed on the prevalence for human echinococcosis using the inverse distance method. Further the prevalence for human CE and AE was analyzed. The global spatial autocorrelation showed that positive spatial autocorrelation and aggregation distribution were present rather than random distribution (Table 5). Local Indicators of Spatial Association (LISA) for human CE significance cluster map showed the presence of two kinds of clusters: high-high and low-high clusters. High-high clusters were mainly distributed in Tibet, western Qinghai, and the Ganzi Tibetan Autonomous Prefecture in Sichuan, with a positive correlation in spatial distribution (Fig 4). The local spatial autocorrelation analysis for human CE showed that the 45 "high-high" gathering areas were mainly

**Table 3. Classification of the prevalence rate for human CE in China, 2018.**

| Province/ Autonomous region | Total number of. counties | $P \geq 100/10000$ | | $10/10000 \leq P < 100/10000$ | | $0 < P < 10/10000$ | | $P = 0$ | |
|---|---|---|---|---|---|---|---|---|---|
| | | Number of counties | constituent ratio (%) | Number of counties | Constituent ratio (%) | Number of counties | Constituent ratio (%) | Nunmber of counties | Constituent ratio (%) |
| Inner Mongolia | 26 | 0 | 0.0 | 1 | 3.8 | 11 | 42.3 | 14 | 53.8 |
| Sichuan | 35 | 3 | 8.6 | 14 | 40.0 | 17 | 48.6 | 1 | 2.9 |
| Yunnan | 24 | 0 | 0.0 | 0 | 0.0 | 8 | 33.3 | 16 | 66.7 |
| Tibet | 74 | 14 | 18.9 | 52 | 70.3 | 7 | 9.5 | 1 | 1.4 |
| Shaanxi | 2 | 0 | 0.0 | 0 | 0.0 | 1 | 50.0 | 1 | 50.0 |
| Gansu | 57 | 0 | 0.0 | 5 | 8.8 | 47 | 82.5 | 5 | 8.8 |
| Qinghai | 39 | 4 | 10.3 | 18 | 46.2 | 17 | 43.6 | 0 | 0.0 |
| Ningxia | 19 | 0 | 0.0 | 2 | 10.5 | 17 | 89.5 | 0 | 0.0 |
| Xinjiang | 81 | 0 | 0.0 | 6 | 7.4 | 71 | 87.7 | 4 | 4.9 |
| Xinjiang Production and Construction Crops | 13 | 0 | 0.0 | 0 | 0.0 | 11 | 84.6 | 2 | 15.4 |
| Total | 370 | 21 | 5.7 | 98 | 26.5 | 207 | 55.9 | 44 | 11.9 |

*P*: prevalence rate

located in the counties of Ngamring, Baqeen, Bainang, Bangoin, Biru, Chagyab, Dinggyee, Tingri, Gar, Geerzee, Gamba, Gee'gyai, Gongbo'gyamda, Konjo, Gyirong, Jiali (Lhari), Kangmar, Nang, Nagarzee, Nyima, Nyalam, Burang, Nagqu, Rutog, Saga, Xigazee, Xainza, Shuanghu, Sog, Xaitongmoin, Yadong (Chomo) and Zhongba in Tibet; Shiqu, Seertar, Deegee, Baiyu and Xinlong (Nyagrong) in Sichuan; Tarlag, Baima, Chindu, Gadee, Jigzhi, Madoi, Maqeen and Yushu in Qinghai (Fig 4). Low-high clusters were distributed in Coqeen county in Tibet, with a negative correlation in spatial distribution (Fig 4). The LISA of human AE indicated the existence of high-high clusters only. These 10 "high-high" gathering areas for AE were predominantly located in the counties of Shiqu, Seertar, Baiyu in Sichuan and Tarlag, Baima, Chindu, Gadee, Jigzhi, Madoi and Maqeen in Qinghai (Fig 5).

### 3.4. Identification of clusters for human echinococcosis

Spatial scan statistics were performed independently for human CE and AE in the 370 endemic counties using SaTScan. The most likely clusters for human CE were found in the north of the Tibet Autonomous Region, with Amdo as the centre of a radius of 888.58 km, covering 120 epidemic counties in the Qinghai-Tibet Plateau. The relative risk (RR) value was 21 and involved about 5.3 million exposed people (Fig 6). A set of 10 secondary clusters were identified (Fig 6). They corresponded to 1 to 3 counties at the most and were thus more epidemic foci than to epidemic areas. Risks of transmission in those endemic counties, as indicated by the RR value, are shown in Table 6 and Fig 6. With respect to human AE, the most likely clusters were found in the Tarlag county of the Qinghai province (Fig 7). This cluster displayed a radius of 888.58 km, covering 15 epidemic counties, with a RR of up to 305.82. This is a very serious situation. Three secondary clusters were also identified corresponding to an epidemic area with a medium epidemic degree and 2 epidemic foci (Table 7 and Fig 7).

### 4. Discussion

In this study, spatial autocorrelation and spatial scan statistics were used to systematically characterize the spatial distribution and prevalence for human echinococcosis in each county. This

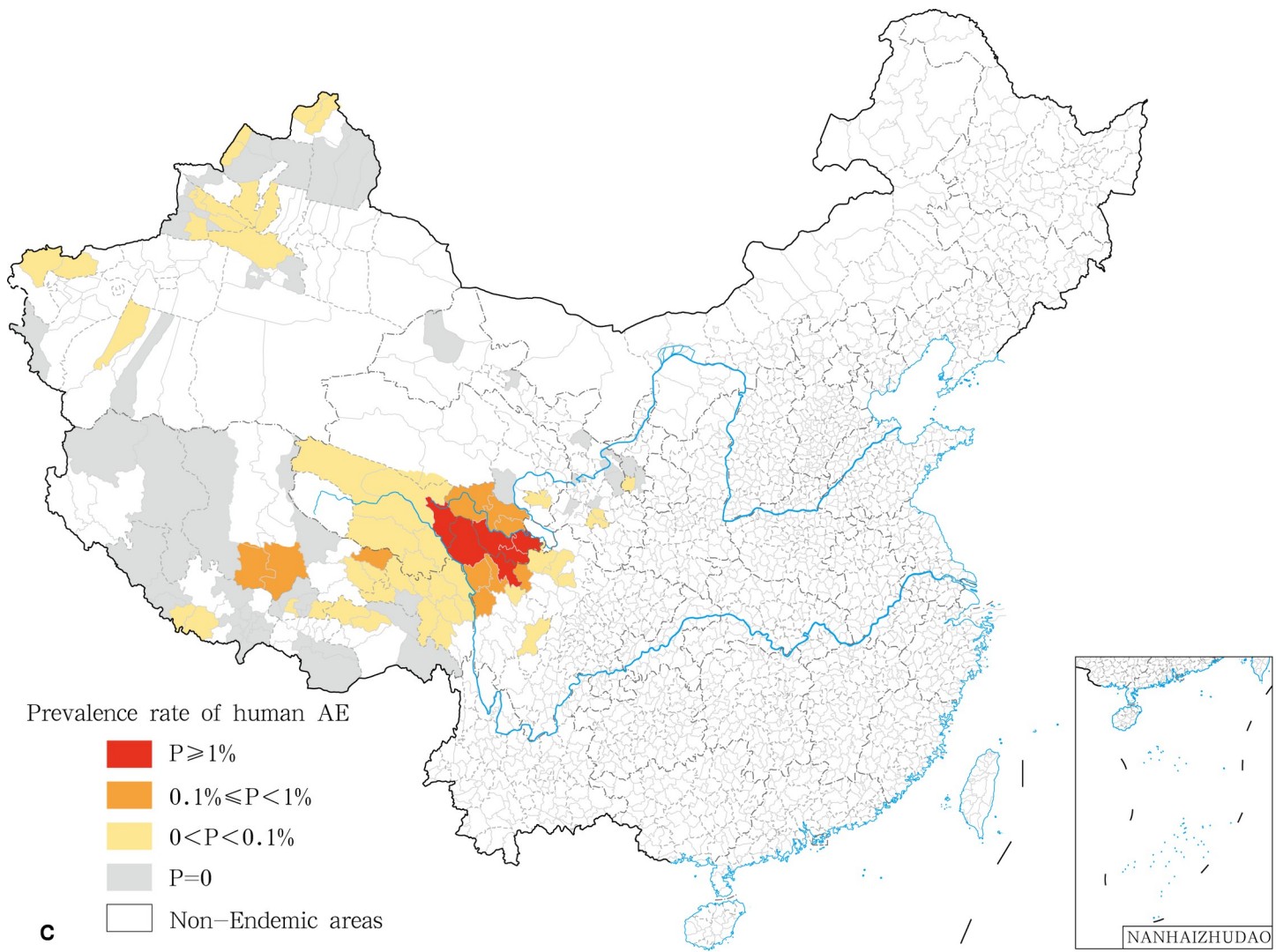

Prevalence rate of human AE

- 🟥 P ≥ 1%
- 🟧 0.1% ≤ P < 1%
- 🟨 0 < P < 0.1%
- ⬜ P = 0
- ☐ Non−Endemic areas

c

NANHAIZHUDAO

**Fig 3. The spatial distribution of human AE in China in 2018.** The base layer is from https://www.webmap.cn/mapDataAction.do?method=forw&resType=5&storeId=
2&storeName=%E5%9B%BD%E5%AE%B6%E5%9F%BA%E7%A1%80%E5%9C%B0%E7%90%86%E4%BF%A1%E6%81%AF%E4%B8%AD%E5%BF%83&fileId=
BA420C422A254198BAA5ABAB9CAAFBC1 with credit to National Catalogue Service For Geographic Information.

is to our knowledge the first time that this level of discrimination and accurracy is reached. Spatial autocorrelation analyses showed statistical significance, revealing that the spatial distribution of human echinococcosis in China displays a non-random spatial clustering pattern. Further spatial autocorrelation analyses of the prevalence of echinococcosis showed that the "hot spots" were concentrated in the Qinghai-Tibet Plateau, mainly in northern Tibet, southwestern Qinghai and Ganzi, and Sichuan. Therefore, echinococcosis patients were mainly concentrated in the Qinghai-Tibet Plateau. Huang *et al* also indicated that the Qinghai-Tibet Plateau region was high-risk area for CE [18]. In addition, several studies have reported that Sichuan, Gansu, Qinghai, and Ninxia were characterized by high prevalence of AE [2]. This could be explained that at such a high altitude, low temperature and high humidity enable eggs to survive longer. A further explanation of the high prevalence observed in the Qinghai-Tibet Plateau is that most of its residents belong to an ethnic minority which adhere to traditional ways of production and life and religious practices. In Tibetan areas particularly, where cattle

**Table 4. Classification of the prevalence rate for human AE in China, 2018.**

| Province/ Autonomous region | Total number of. counties | P ≥ 100/10000 | | 10/10000 ≤ P < 100/10000 | | 0 < P < 10/10000 | | P = 0 | |
|---|---|---|---|---|---|---|---|---|---|
| | | Number of counties | constituent ratio (%) | Number of counties | Constituent ratio (%) | Number of counties | Constituent ratio (%) | Nunmber of counties | Constituent ratio (%) |
| Inner Mongolia | 0 | 0 | 0 | 0 | 0 | 0 | 0 | 0 | 0 |
| Sichuan | 11 | 2 | 18.2 | 4 | 36.4 | 5 | 45.5 | 0 | 0 |
| Yunnan | 0 | 0 | 0 | 0 | 0 | 0 | 0 | 0 | 0 |
| Tibet | 47 | 0 | 0 | 3 | 6.4 | 17 | 36.2 | 27 | 57.4 |
| Shaanxi | 0 | 0 | 0 | 0 | 0 | 0 | 0 | 0 | 0 |
| Gansu | 10 | 0 | 0 | 0 | 0 | 2 | 20 | 8 | 80 |
| Qinghai | 14 | 4 | 28.6 | 3 | 21.4 | 6 | 42.9 | 1 | 7.1 |
| Ningxia | 3 | 0 | 0 | 0 | 0 | 1 | 33.3 | 2 | 66.7 |
| Xinjiang | 30 | 0 | 0 | 0 | 0 | 15 | 50 | 15 | 50 |
| Xinjiang Production and Construction Crops | 0 | 0 | 0 | 0 | 0 | 0 | 0 | 0 | 0 |
| Total | 115 | 6 | 5.2 | 10 | 8.7 | 46 | 40 | 53 | 46.1 |

*P*: prevalence rate

and sheep are mainly slaughtered in families, stray dogs are present in high number and are fed with *Echinococcus*-infected remains [19,20]. Dogs are definitive hosts for *E. granulosus senso lato* and *E. multilocularis* in China [8,21]. *E. granulosus senso lato* and *E. multilocularis* have different intermediate hosts [21]. *Echinococcus* parasites need two mammalian hosts to complete their life cycle [7]. Transmission of *E. granulosus s.l.* occurs between dogs and live-stock, while *E. multilocularis* transmission occurs between stray dogs or foxes as definitive hosts and rodents as intermediate hosts [21]. Humans are aberrant intermediate hosts and they are infected by ingesting *E. multilocularis* eggs [22]. The involvement of wildlife in *E. multilocularis* transmission cycle is not clear and further researches are needed to investigate this issue. However, the spatial distribution of the prevalence of human echinococcosis directly reflects the range and degree of the corresponding transmission cycle of CE and AE. It can thus guide prevention and control activities, such as strengthening the control of stray dogs in the epidemic areas of AE, carrying out control measures of intermediate host animal density and monitoring the infection status. Health education and people awareness should also be considered. According to the different epidemic degrees of CE and the local control capacity, the frequency of deworming should be determined to achieve the purpose of scientific guidance.

There is a focal spatial distribution for echinococcosis infections, with defined areas at high risk for parasite transmission between definitive and intermediate hosts, where the prevalence or incidence of human echinococcosis may be higher than in surrounding areas [7]. Humans may be accidentally infected through food or water consumption. Dogs and livestocks can contaminate rivers and the environment [23–25]. The Buddhist doctrine applied by local

**Table 5. Spatial autocorrelation global Moran's *I* analysis for the prevalence of echinococcosis.**

| The Prevalence type | Moran's*I* Index | Expected value | Variance | Z-value | P-value |
|---|---|---|---|---|---|
| Total prevalence | 0.276838 | -0.002817 | 0.000161 | 22.055141 | <0.01 |
| Prevalence of CE | 0.259658 | -0.002817 | 0.000168 | 20.272646 | <0.01 |
| Prevalence of AE | 0.322806 | -0.008696 | 0.000784 | 11.843104 | <0.01 |

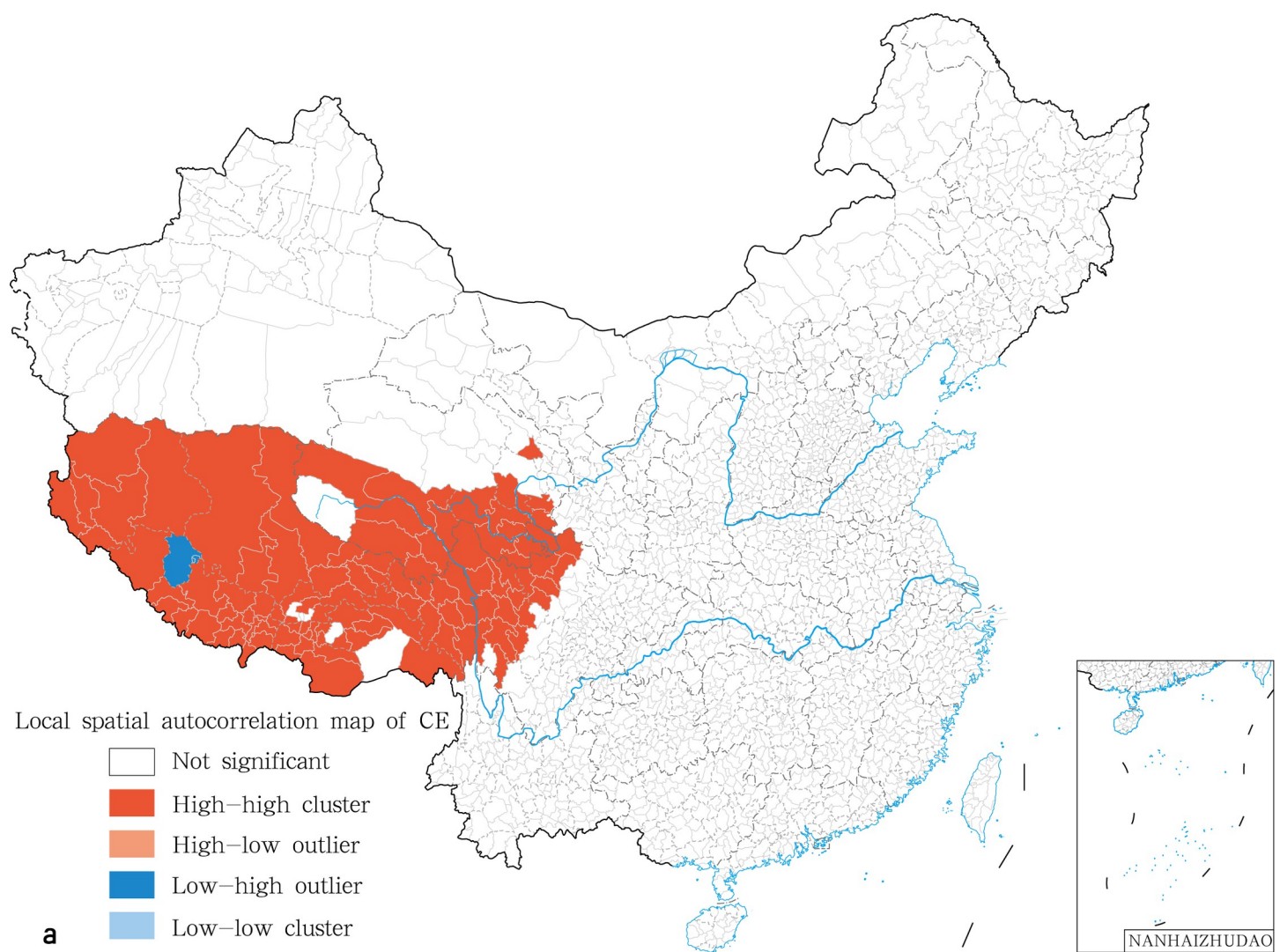

**Fig 4. Local spatial autocorrelation map of human CE in China in 2018.** The base layer is from https://www.webmap.cn/mapDataAction.do?method=forw&resType=5&storeId=2&storeName=%E5%9B%BD%E5%AE%B6%E5%9F%BA%E7%A1%80%E5%9C%B0%E7%90%86%E4%BF%A1%E6%81%AF%E4%B8%AD%E5%BF%83&fileId=BA420C422A254198BAA5ABAB9CAAFBC1 with credit to National Catalogue Service For Geographic Information.

pastoral communities considers that old livestock should die naturally. There is also a practice of unrestricted disposal of animal viscera. These have been identified as risk factors for human CE and AE in Tibetan communities [26]. Furthermore, Tibetan herdsmen families keep at least one dog and pastoralists and Buddhist monks always have large numbers of ownerless stray dogs to stay [27]. Previous investigations have shown that owned dogs were the main transmission factor for both CE and AE on the Qinghai-Tibet Plateau [28,29]. Other components that maintain the high level of transmission in the Qinghai-Tibet Plateau are the lack of preventive measures, limited awareness of disease transmission, and traditional way of life, exposing people to the parasite [30,31].

The results of spatial autocorrelaction and spatial scan cluster analysis confirmed previous studies which suggested a highest rate of human AE in the Qinghai-Tibetan Plateau [27]. This work also demonstrated that Tibet, Xinjiang Uygur Autonomous Region, Ningxia Hui

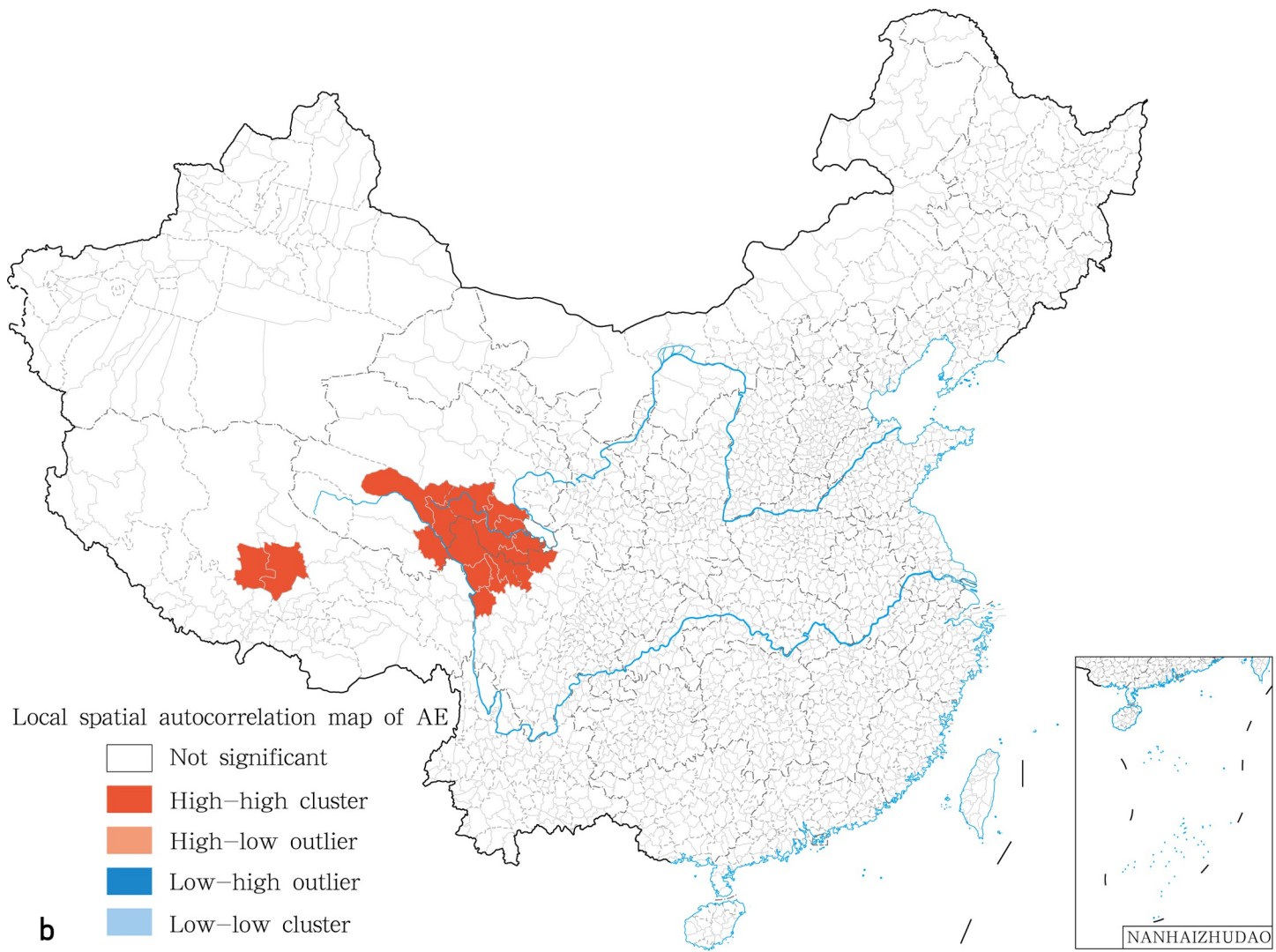

**Fig 5. Local spatial autocorrelation map of human AE in China in 2018.** The base layer is from https://www.webmap.cn/mapDataAction.do?method=forw&resType=
5&storeId=2&storeName=%E5%9B%BD%E5%AE%B6%E5%9F%BA%E7%A1%80%E5%9C%B0%E7%90%86%E4%BF%A1%E6%81%AF%E4%B8%AD%E5%BF%
83&fileId=BA420C422A254198BAA5ABAB9CAAFBC1 with credit to National Catalogue Service For Geographic Information.

Autonomous Region (NHAR) as well as Sichuan and Qinghai Provinces have much higher prevalence for CE [27].

Echinococcosis is a zoonotic parasitic disease and the prevalence is only one of the indicators that reflects the disease endemicity. It mainly indicates the clinical burden and demand for treatment, but cannot reflect the exact risk of transmission which is affected by the period of latency and treatment interventions. Many parameters are involved in parasite transmission, including environmental influences, climate change, anthropogenic environmental factors, and landscape [7]. The relationship between the risk of echinococcosis infection and environmental factors can influence the rate of development of the parasite [32]. In addition, definitive and intermediate hosts infection rates are also important indicators for the presence of parasites [7,33]. It is therefore important to improve the quality of surveillance, optimize its use and take into account relevant indicators (such as dog infection rate and intermediate host prevalence rate). Dogs have close relationship with humans and represent a major source of

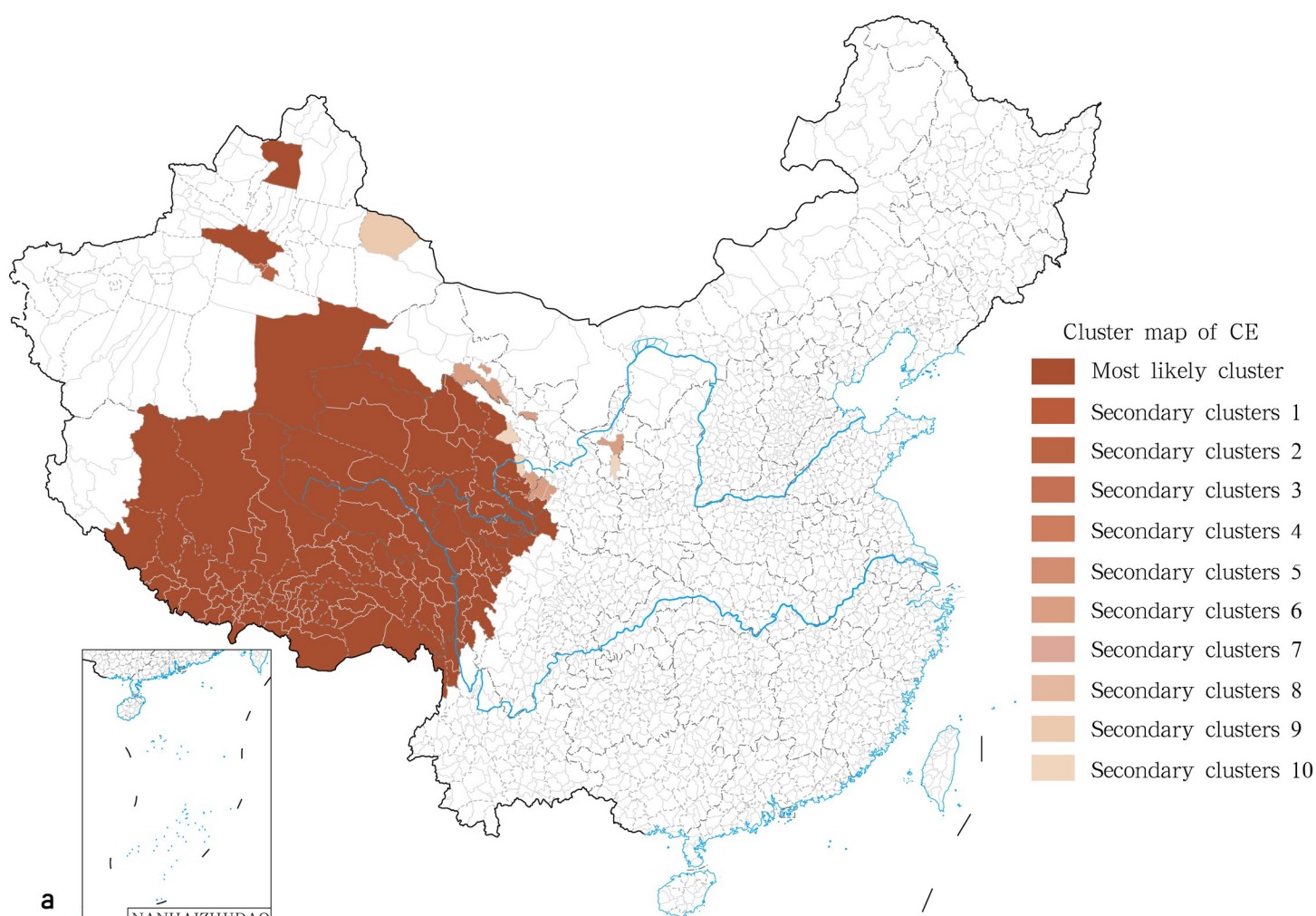

**Fig 6. The spatial aggregation analysis of human CE in China in 2018.** The base layer is from https://www.webmap.cn/mapDataAction.do?method=forw&resType= 5&storeId=2&storeName=%E5%9B%BD%E5%AE%B6%E5%9F%BA%E7%A1%80%E5%9C%B0%E7%90%86%E4%BF%A1%E6%81%AF%E4%B8%AD%E5%BF% 83&fileId=BA420C422A254198BAA5ABAB9CAAFBC1 with credit to National Catalogue Service For Geographic Information.

infection [34–36]. Therefore, strengthening dogs monitoring is key to reduce the risk of infection [37]. Wild canids should not be neglected as they are important wild hosts of *E. granulosus*, in particularl in areas where they are in contacts with humans and domestic animal [38]. Indeed, 47% of domestic dogs may have direct contact with feral dogs [37]. Furthermore, the parasite can be succesfully transmitted between wild and domestic hosts [39]. Meanwhile, it is thus essential to also monitor wild canids. Surveillance should not only monitor the epidemic situation of echinococcosis but can also lead to efficient control measures [40]. However, although essential, a comprehensive surveillance strategy is complex and difficult to implement because of the involvement of many hosts in the life cycle of *E.granulosus* [40]. Permanent surveillance and data collection systems must be implemented to assess the epidemic trend by continuously recording changes in prevalence and other indicators. Joint efforts must be engaged to improve and strengthen the effectiveness of echinococcosis control in the Qinghai-Tibetan Plateau and other epidemic regions. A fixed-point surveillance of echinococcosis in the 370 endemic counties was thus initiated.

**Table 6. Spatial clustering analysis for human CE in 2018.**

| Cluster | The center point | | Scope | | exposed population | Radius (km) | Expected cases | Number of cases | RR | LLR | P-value |
|---|---|---|---|---|---|---|---|---|---|---|---|
| | Latitude | longitude | center county | Number of counties | | | | | | | |
| Most likely cluster | 32.249600 N | 92.625964 E | Amdo | 120 | 5289808 | 888.58 | 4099 | 25125 | 21.09 | 34908.03 | <0.01 |
| Secondary cluster1 | 33.663502 N | 102.883471 E | Zoigee | 1 | 67510 | 0 | 52 | 411 | 7.94 | 490.44 | <0.01 |
| Secondary cluster2 | 42.806400 N | 85.202339 E | Hejing | 1 | 102817 | 0 | 80 | 357 | 4.52 | 259.25 | <0.01 |
| Secondary cluster3 | 46.247299 N | 86.211422 E | Hoboksar | 1 | 55056 | 0 | 43 | 163 | 3.83 | 98.37 | <0.01 |
| Secondary cluster4 | 42.020001 N | 86.301763 E | Yanqi | 2 | 168625 | 51.75 | 111 | 273 | 2.46 | 83.40 | <0.01 |
| Secondary cluster5 | 36.992901 N | 106.288186 E | Tongxin | 1 | 253998 | 0 | 197 | 344 | 1.76 | 45.21 | <0.01 |
| Secondary cluster6 | 35.018601 N | 102.495084 E | Tongren | 3 | 277495 | 59.14 | 215 | 330 | 1.54 | 26.57 | <0.01 |
| Secondary cluster7 | 38.926201 N | 99.300190 E | Sunan | 1 | 38000 | 0 | 29 | 67 | 2.28 | 17.55 | <0.01 |
| Secondary cluster8 | 36.208500 N | 106.242491 E | Yuanzhou | 1 | 354321 | 0 | 219 | 307 | 1.41 | 15.82 | <0.01 |
| Secondary cluster9 | 44.255501 N | 92.819469 E | Barkol | 1 | 80023 | 0 | 62 | 107 | 1.73 | 13.41 | <0.01 |
| Secondary cluster10 | 37.063000 N | 100.756611 E | Haiyan | 1 | 36029 | 0 | 28 | 54 | 1.94 | 9.55 | <0.05 |

RR: Relative risk

LLR: Log likelihood ratio

Western China is the main epidemic area of echinococcosis, whith negative consequences on economic development and health [6]. However, the disease burden is not evenly distributed. Most of the areas where echinococcosis is endemic displayed a relatively low prevalence except the Qinghai-Tibet Plateau which is under serious echinococcosis epidemic pressure. This region should thus be prioritarily considered for the implementation of disease management procedures and policies. Blocking the tranmission of human echinococcosis in the Qinghai-Tibet Plateau appears to be a sanitary priority since this disease not only takes a toll on human lives but also affects the well-being in an already resource-poor area facIltaing thus the extension of other diseases. Beside the Qinghai-Tibet Plateau, some counties were shown to be severely endemic in the Xinjiang Uygur Autonomous Region, Qinghai, Gansu, and Sichuan. These clusters represent a serious risk and must entail continuous attention and upscaled control measures against echinococcosis.

## 5. Conclusion

This work assessed the first nationwide spatial distribution of human echinococcosis. The study identified counties at high-risk of human echinococcosis and showed the presence of "hot spots" which need to be rationally addressed as a priority. Collectively, this study provides a basis for preventive actions and management policy in order to reduce the impact of human echinococcosis and improve its control in China.

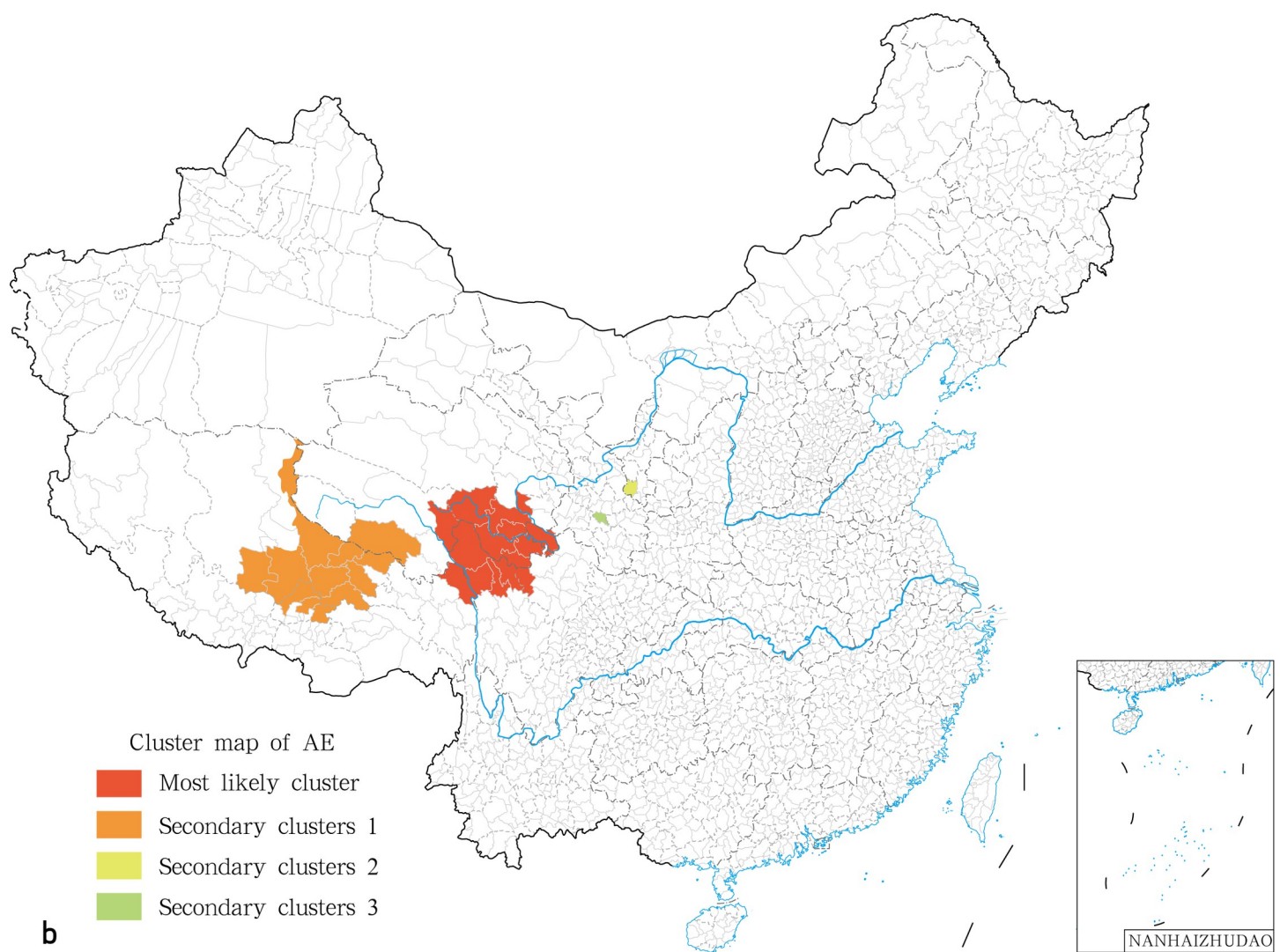

**Fig 7. The spatial aggregation analysis of human AE in China in 2018.** The base layer is from https://www.webmap.cn/mapDataAction.do?method=forw&resType= 5&storeId=2&storeName=%E5%9B%BD%E5%AE%B6%E5%9F%BA%E7%A1%80%E5%9C%B0%E7%90%86%E4%BF%A1%E6%81%AF%E4%B8%AD%E5%BF% 83&fileId=BA420C422A254198BAA5ABAB9CAAFBC1 with credit to National Catalogue Service For Geographic Information.

**Table 7. Spatial clustering analysis for human AE in 2018.**

| Cluster | The center point | | Scope | | exposed population | Radius (km) | Expected cases | Number of cases | RR | LLR | P-value |
|---|---|---|---|---|---|---|---|---|---|---|---|
| | Latitude | longitude | center county | Number of counties | | | | | | | |
| Most likely cluster | 33.479301 N | 99.406402 E | Tarlag | 15 | 774723 | 227.66 | 190 | 9063 | 305.82 | 31968.09 | <0.01 |
| Secondary cluster1 | 32.436901 N | 91.274500 E | Amdo | 12 | 574578 | 293.80 | 141 | 660 | 4.92 | 512.00 | <0.01 |
| Secondary cluster2 | 35.936600 N | 105.719195 E | Xiji | 1 | 408199 | 0.00 | 100 | 185 | 1.86 | 28.90 | <0.01 |
| Secondary cluster3 | 34.7252998 N | 104.3562251 E | Zhang | 1 | 57300 | 0.00 | 14 | 41 | 2.92 | 16.94 | <0.01 |

## Acknowledgments

Since the national echinococcosis control project was launched in 2006, it has been supported by peers in all epidemic areas. We sincerely thank all the participants who participated in the prevention and control activities in 370 echinococcosis endemic counties in 9 epidemic provinces (plus Xinjiang production and Construction Corps).

## Author Contributions

**Conceptualization:** Li-Ying Wang, Wei-Ping Wu, Ning Xiao, Xiao-Nong Zhou.

**Data curation:** Li-Ying Wang, Xiao-Nong Zhou.

**Formal analysis:** Li-Ying Wang, Min Qin, Ze-Hang Liu.

**Funding acquisition:** Li-Ying Wang.

**Investigation:** Li-Ying Wang.

**Methodology:** Li-Ying Wang, Xiao-Nong Zhou, Laurent Gavotte, Roger Frutos.

**Project administration:** Li-Ying Wang.

**Resources:** Li-Ying Wang, Xiao-Nong Zhou.

**Software:** Li-Ying Wang, Laurent Gavotte, Roger Frutos.

**Supervision:** Sylvie Manguin, Laurent Gavotte, Roger Frutos.

**Validation:** Li-Ying Wang, Laurent Gavotte, Roger Frutos.

**Visualization:** Li-Ying Wang, Min Qin, Xiao-Nong Zhou.

**Writing – original draft:** Li-Ying Wang, Xiao-Nong Zhou, Roger Frutos.

**Writing – review & editing:** Xiao-Nong Zhou, Sylvie Manguin, Laurent Gavotte, Roger Frutos.

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
