## [Decision Letter · Decision Letter 0]

2 Jul 2021

Dear Prof. Zhou,

Thank you very much for submitting your manuscript "Prevalence and spatial distribution characteristics of human echinococcosis in China" for consideration at PLOS Neglected Tropical Diseases. As with all papers reviewed by the journal, your manuscript was reviewed by members of the editorial board and by several independent reviewers. The reviewers appreciated the attention to an important topic. Based on the reviews, we are likely to accept this manuscript for publication, providing that you modify the manuscript according to the review recommendations. 

Sincerely,

David Joseph Diemert, M.D.

Associate Editor

Mar Siles-Lucas

Deputy Editor

Reviewer's Responses to Questions

**Key Review Criteria Required for Acceptance?**

**Methods**

-Are the objectives of the study clearly articulated with a clear testable hypothesis stated?

-Is the study design appropriate to address the stated objectives?

-Is the population clearly described and appropriate for the hypothesis being tested?

-Is the sample size sufficient to ensure adequate power to address the hypothesis being tested?

-Were correct statistical analysis used to support conclusions?

-Are there concerns about ethical or regulatory requirements being met?

Reviewer #1: In current study, the objectives were clearly articulated with testable background. Study design is appropriate to address the objectives with clearly described study population. In this study, correct statistical analysis is applied to address the prevalence and spatial distribution of human echinococcosis in China. This study met with the related ethical or regulatory requirements.

Reviewer #2: (No Response)

**Results**

-Does the analysis presented match the analysis plan?

-Are the results clearly and completely presented?

-Are the figures (Tables, Images) of sufficient quality for clarity?

Reviewer #1: The results are clearly and completely presented, however, the figures are lack of high resolution.

Reviewer #2: (No Response)

**Conclusions**

-Are the conclusions supported by the data presented?

-Are the limitations of analysis clearly described?

-Do the authors discuss how these data can be helpful to advance our understanding of the topic under study?

-Is public health relevance addressed?

Reviewer #1: The conclusions are supported by the presented data with its own limitations. The authors discussed the potential relevance of current study to public health.

Reviewer #2: (No Response)

**Editorial and Data Presentation Modifications?**

Reviewer #1: (No Response)

Reviewer #2: (No Response)

**Summary and General Comments**

Reviewer #1: Xiao-Nong Zhou and colleagues reported their original finding on prevalence and spatial distribution characteristics of human echinococcosis in China. This paper is informative, well written and may provide a thorough information regarding the prevalence and spatial distribution characteristics of human CE and AE in China. These date would be quite helpful for implementing prevention and control strategy for human echinococcosis and also is important for fighting against echinococcosis.

However, there are still some concerns regarding this paper:

1. Page4, Line 73, “For the year 2018, 47,233”. This presentation is quite misunderstanding, please correct;

2. Page 11, Line 190, “The number of CA cases……”. This should be CE please correct.

3. Page 11 Line 198. In this section, author stated there are still some unclassified cases. Would you please explain why in some counties have unclassified cases? 

4. Page 13, Line 209 “In the national echinococcosis prevention and control” should be written as “National Echinococcosis Prevention and Control”.

5. Page 18, Line 250, “Further analyzed the prevalence of human CE and the prevalence of human AE ” . This sentence should be corrected and whole paper should be revised for English for better reading.

Reviewer #2: In the MS, the authors collected 47,233 echinococcosis cases in 370 counties of China in China and analyzed the prevalence and spatial distribution characteristics. They found that the endemic counties showed spatial positive autocorrelation in globe spatial autocorrelation with two aggregation modes in local spatial autocorrelation, namely high-high and low-high aggregation areas. Two spatial clusters were revealed by spatial scanning analysis. They confirmed that The Qinghai-Tibet Plateau is the "hot spot" area of human echinococcosis in China. The results are of great importance for prevention and control of echinococcosis in china and are attractive for researches in the field. There are some suggestions for improvement of the MS:

1. The ms is suggested to be reviewed by people whose native language is English.

2. In Abstract, the part of methods is too brief. For example, the distribution of 370 counties should be described. SPSS 21.0 and ArcGIS 10.1 were used to obtain the prevalence rate of AE and CE. Chi-square test and Exact probability method were used to get the message of ?......The overall situation or prevalance rate of echinococcosis in China should be presented in the part of results as well as in Conclusion.

3. P5 line 103, Xinjiang should be replaced by Xinjiang Uygur Autonomous Region

4. P15 Table 2, please check the meaning of P, p value or prevalence rate?

5. P16 Table3, the meaning of P should showed below the table as well as Table 4

6. P18 line 249-250, “the prevalence of human CE and the prevalence of human AE” can be replaced by “the prevalence of human CE and AE” as well as P24 line 309, “ways of production, way of life” to “ways of production, and life and religious…”

7. P25 line 324, “ownd dogs” should be “owned dogs”.

8. P26 line 339-340 not only the gradient was west-east oriented but no north-south variation was observed. What the meaning? Plz check

9. In Discussion, the difference of prevalence rate and spatial distribution between AE and CE should be discussed and analyzed.

PLOS authors have the option to publish the peer review history of their article (what does this mean?). If published, this will include your full peer review and any attached files.

Reviewer #1: Yes: Tuerhongjiang Tuxun

Reviewer #2: No

Figure Files:

Data Requirements:

Reproducibility:

References

---

## [Decision Letter · Decision Letter 1]

2 Oct 2021

Dear Prof. Zhou,

Thank you very much for submitting your manuscript "Prevalence and spatial distribution characteristics of human echinococcosis in China" for consideration at PLOS Neglected Tropical Diseases. As with all papers reviewed by the journal, your manuscript was reviewed by members of the editorial board and by several independent reviewers. The reviewers appreciated the attention to an important topic. Based on the reviews, we are likely to accept this manuscript for publication, providing that you modify the manuscript according to the review recommendations. 

Please note that in contrast to Reviewer 1's comment, the correct spelling is, "reinforcement".

Sincerely,

David Joseph Diemert, M.D.

Associate Editor

Mar Siles-Lucas

Deputy Editor

Please note that in contrast to Reviewer 1's comment, the correct spelling is, "reinforcement".

Reviewer's Responses to Questions

**Key Review Criteria Required for Acceptance?**

**Methods**

-Are the objectives of the study clearly articulated with a clear testable hypothesis stated?

-Is the study design appropriate to address the stated objectives?

-Is the population clearly described and appropriate for the hypothesis being tested?

-Is the sample size sufficient to ensure adequate power to address the hypothesis being tested?

-Were correct statistical analysis used to support conclusions?

-Are there concerns about ethical or regulatory requirements being met?

Reviewer #1: The objectives of the study is clearly articulated with a clear testable hypothesis;

The study design is appropriate to address the stated objectives;

The population is clearly described and appropriate for the hypothesis being tested;

The sample size is sufficient to ensure adequate power to address the hypothesis being tested;

The correct statistical analysis were used to support conclusions;

No concerns exist about ethical or regulatory requirements being met.

**Results**

-Does the analysis presented match the analysis plan?

-Are the results clearly and completely presented?

-Are the figures (Tables, Images) of sufficient quality for clarity?

Reviewer #1: The analysis presented matchs the analysis plan;

The results are clearly and completely presented;

The figures are presented with sufficient quality for clarity.

**Conclusions**

-Are the conclusions supported by the data presented?

-Are the limitations of analysis clearly described?

-Do the authors discuss how these data can be helpful to advance our understanding of the topic under study?

-Is public health relevance addressed?

Reviewer #1: The conclusions are supported by the data presented;

The limitations of analysis are clearly described;

The authors discuss how these data can be helpful to advance our understanding of the topic under study;

Public health relevance is addressed.

**Editorial and Data Presentation Modifications?**

Reviewer #1: 1. Page 4 Line 73, "renforcement" should be written as "reinfoecement";

2. Page 5 Line 102 "They heavily impair the patients, especially AE, with a mortality rate of about 90% in the past ten years" . Please confime ,the mortaliry is about 90% in AE without any intervention.

3. Page 23 Line 303 "explaned" shoud be written as "explained".

**Summary and General Comments**

Reviewer #1: 1. Page 4 Line 73, "renforcement" should be written as "reinfoecement";

2. Page 5 Line 102 "They heavily impair the patients, especially AE, with a mortality rate of about 90% in the past ten years" . Please confime ,the mortaliry is about 90% in AE without any intervention.

3. Page 23 Line 303 "explaned" shoud be written as "explained".

PLOS authors have the option to publish the peer review history of their article (what does this mean?). If published, this will include your full peer review and any attached files.

Reviewer #1: Yes: Tuerhongjiang Tuxun

Figure Files:

Data Requirements:

Reproducibility:

References

---

## [Editor Report · Decision Letter 2]

15 Nov 2021

Dear Prof. Zhou,

We are pleased to inform you that your manuscript 'Prevalence and spatial distribution characteristics of human echinococcosis in China' has been provisionally accepted for publication in PLOS Neglected Tropical Diseases.

Best regards,

David Joseph Diemert, M.D.

Associate Editor

Mar Siles-Lucas

Deputy Editor

---

## [Editor Report · Acceptance letter]

20 Dec 2021

Dear Prof. Zhou,

We are delighted to inform you that your manuscript, "Prevalence and spatial distribution characteristics of human echinococcosis in China," has been formally accepted for publication in PLOS Neglected Tropical Diseases.

Best regards,

Shaden Kamhawi

co-Editor-in-Chief

Paul Brindley

co-Editor-in-Chief
